# Understanding Chile Pepper Consumers' Preferences: A Discrete Choice Experiment

Jay Lillywhite [1],* and Chadelle Robinson [2]

1 College of Agricultural, Consumer, and Environmental Sciences, New Mexico State University, Las Cruces, NM 88003, USA

2 Department of Agricultural Economics and Agricultural Business, College of Agricultural, Consumer, and Environmental Sciences, New Mexico State University, Las Cruces, NM 88003, USA; chadelle@nmsu.edu

* Correspondence: lillywhi@nmsu.edu

**Abstract:** U.S. per-capita chile consumption and foreign imports have increased over the last twenty years while domestic production has fallen. To maintain market share, U.S. chile producers must increase crop revenues and/or decrease production expenses. A better understanding of U.S. consumer preferences relative to chile attributes can provide direction for U.S. chile producers. This paper utilizes a discrete choice experiment within an online survey to gain insights into long-green chile pepper attributes desired by consumers. The results suggest that survey participants prefer fresh long-green chile produced in the United States. Participants also preferred milder long-green chile and value quality inspections. Organic production was preferred to hydroponically produced long-green chile, but a statistical difference between organic and other production practices was not observed. Understanding these preferences may allow producers to better position themselves to remain competitive in the long-green chile market.

**Keywords:** spicy peppers; chile; consumer preferences





## 1. Introduction

Spicy peppers from the Capsicum genus, also referred to as chili or chili peppers, chile or chile peppers, spicy peppers, or hot peppers, come from one of five domesticated species, including *Capsicum annum*, *Capsicum chinese*, *Capsicum frutescens*, *Capsicum baccatum*, and *Capsicum pubescens* [1]. A commonly grown spicy pepper is the New Mexico type long-green chile pepper, sometimes referred to as the Anaheim chile. Anaheim chile peppers were originally imported to California from New Mexico by Emilio C. Ortega [2,3]. While the chiles have the same heritage, growing conditions likely impact the taste and heat of the peppers, making the two distinctly different from each other. Chile peppers are believed to have originated from South America and cultivated in Mexico. Indeed, chile peppers are one of the oldest cultivated crops in the Americas [4].

Today, green chile (*Capsicum* spp. and *Pimenta* spp.) is grown in 125 countries, with more than two million hectares (2,055,310 hectares or 5,078,782 acres) producing more than 36 million tonnes (36,286,644 tonnes or 714,271,038 cwt.) in 2021. Leading countries include China, Türkiye, Indonesia, Mexico, and Spain. The leading continent, by far, is Asia (Figure 1). The United States ranked sixteenth in total production area and ninth in production, with more than 500,000 tonnes produced in 2021 [5]. A majority of green chile production in the United States is centered in New Mexico and California, as shown in Figure 2 [6].

An examination of Figure 2 shows that U.S. chile pepper production has decreased over the last 20 or more years. The decline may be attributed to various factors, including increased international trade associated with trade agreements, labor availability, crop returns, and new and alternative crop introductions [7]. In addition to these factors, the

states that produce chile peppers commercially also face significant water issues that may threaten future agricultural production, including chile peppers.

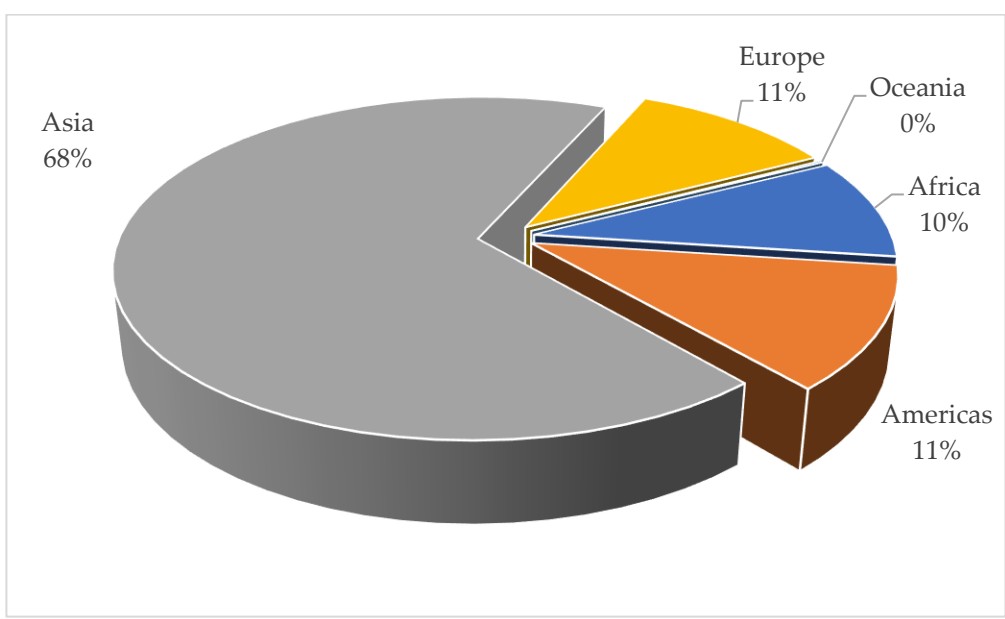

**Figure 1.** Green chiles and peppers (*Capsicum* spp. & *Pimenta* spp.) Production Continents [5].

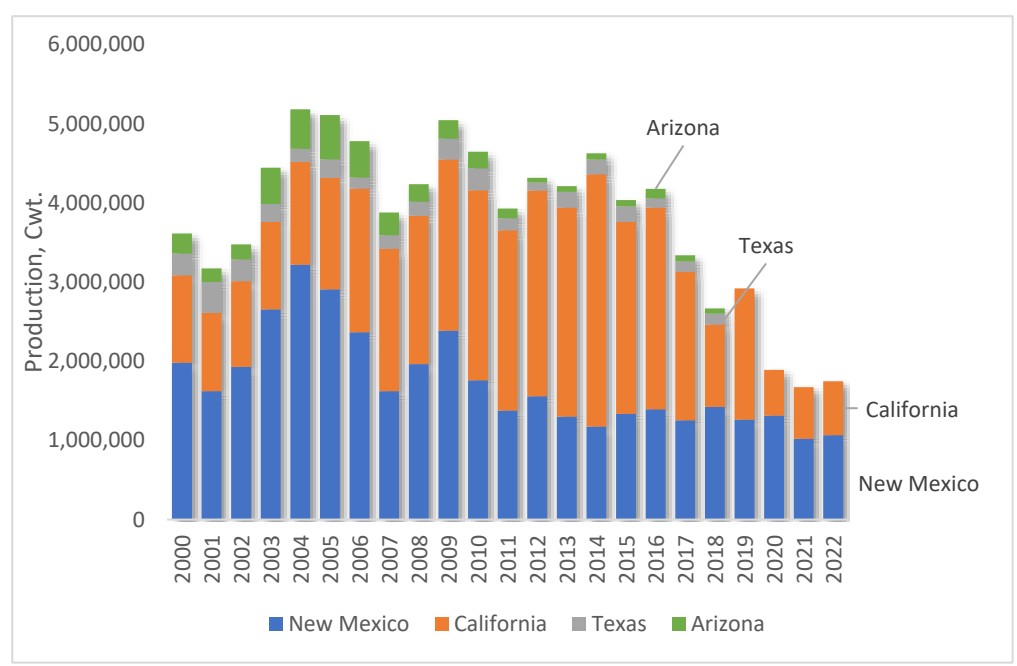

**Figure 2.** Chile pepper production, 2000–2022 [8].

While production in the United States has decreased, worldwide production has increased. Notable examples of increased production between 2000 and 2021 for countries with sizeable production (more than one million tonnes of production in 2021) include Indonesia (277%), Türkiye (109%), China (78%), Mexico (49%), and Spain (60%) [5]. A significant amount of chile products have been imported into the United States. Between 2000 and 2022, imports of chile peppers increased by more than 200%, with the vast majority (more than 98%) coming from Mexico [9].

To survive and thrive, domestic chile pepper producers must be able to increase their crop revenues, decrease crop expenses, or both. One potential way in which producers

may increase their crop revenue is to identify and capitalize on crop characteristics desired by consumers. For example, if consumers desire crops produced using organic production methods, producers may be able to obtain a premium for the crop, increasing their revenues. Or alternatively, from a cost-reduction standpoint, it may be possible to reduce water use, a limited resource in current chile pepper-producing regions, through indoor or hydroponic production systems, if consumers are accepting production changes.

This paper explores chile pepper characteristics demanded by consumers or potential consumers, specifically characteristics for New Mexico-type long-green chile, hereafter referred to as "long-green chile," using data collected from a national online panel survey in 2021. The importance of various chile pepper attributes, including production region, production type, quality certification, pungency, and price, are identified using a discrete choice experiment and analysis. By better understanding consumer preferences for long-green chile, domestic producers may be able to adjust their production and marketing practices align with consumer preferences and maintain market share.

## 2. Materials and Methods

### 2.1. Data

Data in the discrete choice analysis was obtained via a nationwide panel survey conducted in July 2021. Survey participants were segmented into two groups, one group receiving additional information about long-green chile production while the other group was not provided the additional information. This paper used the sample (n = 477) of survey participants who did not receive additional information to avoid bias that could result from learning more about chile production. Invitations to complete the survey hosted on the online survey platform Qualtrics were managed by the online panel management company Cint. Cint is one of a number of online panel management companies, reaching more than 4600 survey panels in more than 130 countries [10]. Eight hundred and fifty-nine panelists, after reading the consent information, agreed to participate in the survey. Four hundred and seventy-seven (477) participants were invited to participate in a discrete choice experiment described in this paper. Table 1 summarizes participant demographics and compares them to those of the U.S. population for 2021, calculated from the U.S. Census American Community Survey [11].

An examination of Table 1 shows that survey demographics generally fit those of the broader population, with some exceptions. For example, the proportion of survey participants from the West Census District was less than those in the population, while the proportion of respondents from the Northeast Census District was higher than the population. Other notable differences between survey and population proportions include age, income, race, and education. Some of the observed differences between the survey demographics and those of the broader population may be attributable to the method of collection. For example, higher-income individuals face higher opportunity costs and thus may be less likely to participate in an online survey. As the data are not necessarily representative of the U.S. population and sampling was not necessarily random, readers should not make inferences from the survey results to the general public. Rather, the results should be considered exploratory in nature, providing important insights but not necessarily conclusive or representative of all consumers.

Survey participants were asked about their spicy pepper consumption, exploring different pepper varieties as well as different processing levels, e.g., fresh, dried, or frozen. Figure 3 summarizes the participants' long-green chile consumption. Approximately one-third of the participants indicated they had consumed canned long-green chile within the last three months. The consumption of frozen and fresh long-green chile in the last three months was reported by approximately one out of five participants.

**Table 1.** Survey and U.S. Population Demographics.

| Demographic | Survey | Census [1] |
|---|---|---|
| Census Region (n = 471) | | |
| Northeast | 23.1% | 17.4% |
| Midwest | 21.7% | 20.8% |
| South | 39.1% | 38.1% |
| West | 16.1% | 23.7% |
| Education (n = 476) | | |
| High School Degree or Less | 26.1% | 38.0% |
| Some College, No Bachelor | 32.6% | 29.5% |
| Bachelor's Degree or Higher | 41.4% | 32.4% |
| Sex (n = 476) | | |
| Female | 50.4% | 50.5% |
| Male | 49.6% | 49.5% |
| Race (n = 477) | | |
| Caucasian (White) | 76.5% | 61.2% |
| African American (Black) | 10.1% | 12.1% |
| Asian | 6.9% | 5.8% |
| Other (including two or more races) | 6.5% | 21.0% |
| Hispanic (n = 471) | 11.3% | 18.8% |
| Income (n = 477) | | |
| Less than $50,000 | 45.1% | 36.5% |
| $50,000 to $99,000 | 37.5% | 29.6% |
| $100,000 or more | 17.4% | 34.0% |
| Age (n = 477) | | |
| Less than 35 years of age | 28.5% | 29.1% |
| 35 to 64 years of age | 48.2% | 49.2% |
| 65 years of age or older | 23.3% | 21.6% |

[1] U.S. Census American Community Survey [11].

**Figure 3.** Survey participant consumption of long-green chile by processing level.

*2.2. Methods*

Discrete choice analysis (DCA) is a commonly used tool used to understand consumer preferences and behavior better [12,13]. It has been successfully used in a variety of disciplines exploring the interaction effects of product characteristics in the decision-making

process. The methodology has been used in numerous applications related to food, food processing, and nutrition.

Discrete choice analysis relies on an "experiment" that attempts to mimic choices faced in real-world situations. Participants are presented with a set (or sets) of different choices and asked to identify the choice they would select if given the opportunity. One advantage of discrete choice analysis and similar or related tools, e.g., conjoint analysis, is that they allow analysts the opportunity to explore products or product formulations that are not necessarily available on the market. Additionally, the tools allow the analyst to understand which product attributes are most desired by participants.

A discrete choice experiment was included in the survey to better understand consumer preferences for long-green chile. Survey participants were presented with a series of three choice sets, each set containing five possible fresh long-green chile choices that included a "would not purchase any of these" option. Variables included in the choices were influenced by previous pepper research (production region, price, quality inspection, and pungency level), summarized in Table 2. In addition, production type was included in the experiment as a means of better understanding participant preferences for alternative, less-commonly used production methods for long-green chile.

**Table 2.** Examples of attributes used in other spicy pepper analyses.

| Attribute | Pepper Type | Author |
|---|---|---|
| Production region | Jalapeño pepper | Toledano et al. [14] |
| Price | Jalapeño pepper | Sánchez-Toledano et al. [15] |
| Quality inspection | Chile pepper | Lillywhite et al. [16] |
| Pungency level | Cayenne and Spicy peppers | Tamba et al. and Lillywhite et al. [17,18] |

Table 3 shows the attributes and attribute levels included in the experiment. Figure 4 illustrates how the choices were presented to participants. As noted above, a majority of the choice experiment attributes were influenced by previous research, with the exception of production types. Options of production types included traditional, organic, indoor soil, and indoor hydroponic. While several of these options are not commonly used in commercial long-green chile production, e.g., indoor production, they were included to explore consumer acceptance of chile grown with alternative, water-saving production techniques.

| **Choice A** | **Choice B** | **Choice C** | **Choice D** | **Choice E** |
|---|---|---|---|---|
| Grown in California | Grown in Florida | Grown in New Mexico | Grown Internationally | |
| $0.89 per pound | $1.09 per pound | $1.29 per pound | $1.49 per pound | Would not purchase any of these |
| Grown traditionally | Grown organically | Grown indoors in soil | Grown indoors hydroponically | |
| Inspected by third party for quality | Inspected by third party for quality | Inspected by third party for quality | Inspected by third party for quality | |
| Mild pungency (Mild heat) | Medium pungency (Medium heat) | High pungency (High heat) | Very high pungency (Very high heat) | |

**Figure 4.** Example choice set provided to participants.

**Table 3.** Attributes and attribute levels used in the discrete choice experiment.

| Attribute | Levels |
| --- | --- |
| Growing region | |
| | Grown in California |
| | Grown in Florida |
| | Grown in New Mexico |
| | Grown Internationally |
| Pungency or heat level | |
| | Mild pungency (heat) |
| | Medium pungency (heat) |
| | High pungency (heat) |
| | Very high pungency (heat) |
| Price | |
| | $0.89 |
| | $1.09 |
| | $1.29 |
| | $1.49 |
| Quality inspection | |
| | Inspected by a third party |
| | Not Inspected by a third party |
| Production practice | |
| | Grown Traditionally |
| | Grown Indoors Hydroponically |
| | Grown Indoors in Soil |
| | Grown Organically |

Theoretically, discrete choice analysis is founded in a random utility framework [12]. In this framework, consumer utility or satisfaction can be broken into two different sources or components, a representative or systematic component ($\gamma$) and a random component ($\varepsilon$). The random component accounts for unobserved differences in consumer preferences [13]. Using this notation, individual i's utility for project j, $U_{ij}$, can be written as

$$U_{ij} = \gamma_{ij} + \varepsilon_{ij} \tag{1}$$

Assuming individuals maximize their utility in that they choose products that give them the most satisfaction, then alternative *j* is chosen if and only if

$$\gamma_{ij} + \varepsilon_{ij} > \gamma_{ik} + \varepsilon_{ik} \ \forall \ k \neq j \tag{2}$$

Rearranging Equation (2) shows that

$$\gamma_{ij} - \gamma_{ik} > \varepsilon_{ik} - \varepsilon_{ij}. \tag{3}$$

As random components represented by $\varepsilon$ are not observable, the analyst must estimate the probability that $\varepsilon_{ik} - \varepsilon_{ij}$ is less than $\gamma_{ij} - \gamma_{ik}$. In order to estimate this probability, parametric and distributional assumptions or specifications must be made. Analysts commonly assume the random components are independent and identically distributed within an extreme-value type distribution. Additionally, the representative components of consumer utility are often considered linear additive functions of product attributes. In this case, the probability that individual *i* selects product *j* can be written as [13].

$$P_{ij} = \frac{e^{\beta_j x_{ij}}}{\sum_{j=1}^{J} e^{\beta_j x_{ij}}} \tag{4}$$

The model parameters, $\beta_j$, within the model, are estimated using maximum likelihood estimation (MLE). The parameter estimates provide a measure of utility associated with individual attributes that can be used to develop measures of the relative importance

participants place on product-specific attributes, e.g., [19–21]. The relative importance of product attribute *i* is calculated as [13]

$$RI = \frac{\left|\beta_i \left(x_i^L - x_i^S\right)\right|}{\left|\sum_{j=1}^{J} \beta_J \left(x_j^L - x_j^S\right)\right|} \tag{5}$$

where $\beta_i$ is the estimated parameter associated with variable $x_i$, $x_i^L$ and $x_i^S$ are the largest observed value and the smallest observed value of variable $x_i$, respectively.

In addition to estimating the relative importance of various product attributes to survey participants, the coefficients in the regression model can be used to approximate participants' willingness to pay for a particular product attribute. Willingness to pay and relative importance derived from discrete choice experiments have been used in numerous food and agriculture-related studies [22–26]. The willingness to pay is approximated by dividing an attribute's estimated parameter coefficient by the negative value of the estimated parameter coefficient for the price variable [13].

$$WTP_i = \frac{\beta_i}{|\beta_p|} \tag{6}$$

where $WTP_i$ is the willingness to pay for attribute *i*, $\beta_i$ is the estimated coefficient for attribute *i*, and $\beta_p$ is the estimated value for the price parameter or coefficient. It should be noted that the willingness to pay approximations calculated using Equation (6) are relatively simple measures and more sophisticated methods can be used [13]. Willingness to pay measures from stated preference models, like those used here, may overstate expected or observed attribute prices [27–30]. As such, it may be more appropriate to use willingness to pay estimates as a relative measure than an absolute measure.

### 3. Results

The software program NLOGIT 6 was used to analyze the data from the discrete choice experiment. A likelihood ratio test was used to compare the "constants-only" model to the main effects or full model (model with all choice-related variables included). The likelihood ratio statistic was 195.43, which was significant at the five-percent level. The discrete choice results, including coefficient estimates, their standard errors, and the corresponding *p*-values, are shown in Table 4.

**Table 4.** Discrete choice multinomial regression results.

| Variable | Coefficient Estimate [1] | Standard Error | Prob |
|---|---|---|---|
| Grown in California | 1.83 | 0.3613 | 0.0000 |
| Grown in Florida | 2.45 | 0.3509 | 0.0000 |
| Grown in New Mexico | 2.23 | 0.2929 | 0.0000 |
| Grown Internationally | 0.89 | 0.3763 | 0.0178 |
| Medium Pungency (Heat) | −1.04 | 0.6314 | 0.0998 |
| High Pungency (Heat) | −0.69 | 0.1252 | 0.0000 |
| Very High Pungency (Heat) | −1.38 | 0.3256 | 0.0000 |
| Price | −1.13 | 0.2268 | 0.0000 |
| Quality Inspected | 1.14 | 0.6296 | 0.0699 |
| Grown Traditionally | −0.39 | 0.5905 | 0.5141 |
| Grown Indoors Hydroponically | −0.36 | 0.1363 | 0.0085 |
| Grown Indoors in Soil | −0.51 | 0.5528 | 0.3589 |

[1] Mild pungency, no quality inspection, and organic production were left out of the model to avoid singularity, i.e., the "dummy variable trap".

Eight of the twelve estimated coefficients were significant at the five-percent level, with an additional two significant at the ten-percent level. The two variables that did not enter

into the regression equation significantly (at the ten-percent level) were associated with production, i.e., growing traditionally and growing indoors in soil. The lack of significance suggests that survey participants did not value these production types more or less than organic production (the variable that was excluded from the equation to avoid singularity).

Using the coefficient estimates shown in Table 4, the relative importance of each attribute was calculated using Equation (5), summing up each level within the attribute. The values indicate the relative importance of each attribute in the decision-making process. Table 5 shows that growing region was the most important attribute in the decision-making process, followed by pungency and growing practice.

**Table 5.** Relative Importance.

| Category | Growing Region |
|---|---|
| Growing region | 54% |
| Pungency | 23% |
| Price | 5% |
| Quality inspection | 8% |
| Growing type | 9% |

Table 6 shows the estimated willingness to pay for each attribute/attribute level, as calculated using Equation (6). Willingness to pay estimates allow researchers to present a vague measure of utility in more familiar units, i.e., dollars. As indicated in the previous section, caution should be taken when examining willingness to pay values, as they can overstate the true value of consumer willingness to pay [27–30]. As such, it may be more appropriate to use the values comparatively, between attributes, rather than as a measure of the true willingness to pay. By construction, the signs of the willingness to pay estimates were generally consistent with the signs of the utility measures, i.e., coefficient estimates and researcher expectations.

**Table 6.** Willingness to Pay.

| Variable | $ |
|---|---|
| Growing region | |
| California | $1.62 |
| Florida | $2.17 |
| New Mexico | $1.98 |
| International | $0.79 |
| Pungency | |
| Medium | −$0.92 |
| Hot | −$0.61 |
| Very Hot | −$1.22 |
| Quality Inspection | $1.01 |
| Growing type | |
| Traditional | −$0.34 |
| Hydroponic | −$0.32 |
| Indoors in soil | −$0.45 |

## 4. Discussion

The regression results were generally consistent with researcher expectations developed through a review of previous research. For example, previous research identified in Table 2 found that consumer preferences are impacted by geographic growing regions and pungency levels. Additionally, inspections are generally viewed by consumers as positive and are utility-increasing. Consistent with the law of demand and previous research, price was found to reduce consumer utility.

If applicable to the population, the results described above bode well for domestic long-green chile producers in that participants indicated a preference for domestically produced

long-green chile. As might be expected, for a general population, survey participant preferences, as a group, were reduced with more pungent long-green chile, suggesting that more mild varieties may be accepted by the general public. Price negatively impacted consumer utility but was the least important of the five long-green chile attributes examined as measured by relative importance.

While results were generally consistent with researcher expectations, there were several exceptions. Two notable exceptions were participants' preference for long-green chile produced in Florida and the lack of a premium for organically grown long-green chile.

While Florida is a leading producer of vegetables nationwide, it does not produce long-green chile commercially as reported by USDA. New Mexico and California have been the nation's leading producers of long-green chile peppers for many years (Figure 2). California is the country's leading state for all vegetable production [31]. New Mexico's chile production has also been publicized via various websites and television shows, e.g., "Hatch Chile". Based on this history and public exposure, it may have been expected that long-green chile produced in California or New Mexico would have produced higher utility levels.

Organic production resulted in higher utility levels for participants, as elicited through the discrete choice experiment, than long-green chile produced indoors hydroponically. To the researchers' knowledge long-green chile is not commercially produced hydroponically, at least in significant volumes, although new technologies and interest in new production processes, e.g., controlled-environment agriculture may influence future production practices. But there was no statistical difference in the participants' preferences between organic and the two remaining soil-production practices, i.e., traditional, and indoor in soil. This may appear counterintuitive in that organic and sometimes greenhouse-produced vegetables tend to sell for a premium in the market. One potential reason for the observation is that a relatively small segment of the population (and presumably the survey sample) is willing and able to pay more for organic produce. For example, the Organic Trade Association reported that organic produce sales continued to increase in 2022 but still only accounted for 15% of total produce sales in the United States [32]. Additionally, a relatively small amount of long-green chile is produced organically. Participants familiar with the pepper variety may have discounted organic production choices for this reason.

## 5. Conclusions

Green chile (*Capsicum* spp. and *Pimenta* spp.) is grown in many countries around the world. The New Mexico-type long-green chile, sometimes referred to as Anaheim chile, is produced primarily in North America, both in the United States and Mexico. Commercial green chile production in the United States is centered in New Mexico and California. While per-capita consumption of chile has increased over the last forty years, domestic production has decreased. The research discussed in this paper has explored consumer uses and preferences for long-green chile. By better understanding consumer preferences, domestic producers may be able to capture a larger share of green chile sales. Additionally, if accepted by consumers, alternative production methods, e.g., indoor hydroponic production, could help alleviate water concerns associated with producing long-green chile.

The exploratory research suggests that participants prefer green chile produced in the United States to international locations. Within the United States, production in Florida was preferred to production in New Mexico and California. As a group, participants also preferred milder green chile compared to more pungent chile. Organic production was preferred to hydroponically produced chile, but a statistical difference between organic and other production practices was not observed. Quality inspection increased participant utility as well.

Domestic long-green chile producers, processors, and other stakeholders may wish to explore ways in which they can capitalize on potential consumer preferences as they related to the attributes described in this paper. For example, identifying long-green chile as having been produced in the United States may resonate with domestic consumers.

Additionally, more mild varieties of long-green chile might be successful in appealing to a larger domestic market. Additional research, as discussed below, should be used to verify the effectiveness of these potential actions.

*Limitations and Further Research*

Several limitations associated with the research should be noted. First, the analysis used a discrete choice experiment to elicit survey participant preferences for long-green chile. Stated preference models are subject to hypothetical bias where participants may indicate preferences that are not validated in their actual behavior, i.e., revealed preference [33].

Some results, discussed above, were inconsistent with previous research or researcher expectations, specifically preferences for growing regions within the United States and production types. These results, or the inconsistency of results relative to previous research or expectations, merit additional research to better understand the reasons behind the findings. Future research could explore these two areas, geographic and production preferences, in more detail, asking participants more directed questions related to the two areas. Potentially qualitative analyses could be conducted via use of open-ended questions or other methods, e.g., focus groups or in-depth-interviews.

Finally, the analysis has focused on participant preferences as a whole. Further research could use more sophisticated methods to elicit preference differences in individuals or groups of individuals. For example, individual-specific variables could be included in the main effects model used in this paper. Alternatively, a generalized multinomial logit model could be developed that would allow for preference heterogeneity among survey participants.

**Author Contributions:** Conceptualization, J.L.; methodology, J.L.; software, J.L.; validation and formal analysis, J.L.; writing—original draft preparation, J.L. and C.R.; writing—review and editing, C.R. and J.L.; visualization, J.L. and C.R.; supervision, J.L.; project administration, J.L. All authors have read and agreed to the published version of the manuscript.

**Funding:** Funding support provided, in part, by the New Mexico State University Agricultural Experiment Station.

**Institutional Review Board Statement:** Institutional Review Board (IRB) approval was received to obtain data from human subjects for this research project. IRB Project #21833.

**Data Availability Statement:** Data are not publicly available per IRB application and participation informed consent.

**Acknowledgments:** The authors wish to thank Amber Montano and Sunshine Tso for their work on this project.

**Conflicts of Interest:** Jay Lillywhite and Chadelle Robinson both have previously received grants from the New Mexico Chile Association and have spoken at industry association meetings. No personal remuneration has been received.

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
