# Peer review of "Understanding Chile Pepper Consumers’ Preferences: A Discrete Choice Experiment"

_agriculture, doi:10.3390/agriculture13091792_

Round 1

Reviewer 1 Report

There are intriguing results regarding organic production over hydroponically produced chiles, but more information about the reasons for this preference would be appreciated. It suggests that further investigation may be needed in this area, since there is no statistical difference between organic and other production practices. This study highlights how domestic producers can leverage consumer preference for fresh green chile made in the U.S. in their marketing and branding efforts by finding that survey participants prefer fresh green chile made in the U.S. In addition, consumers value milder green chile and quality inspections, which offers U.S. chile producers valuable insights that can assist them in tailoring their products accordingly.

It would be helpful if the abstract contained more data-oriented information and valuable findings.

In addition to presenting the data, the paper will also provide more actionable recommendations based on the findings to help U.S. chile producers making the most of the findings.

There is a need to write down the materials and methods in a section-by-section manner.

I would like to suggest you look into the citation issue, as it is not being used in factual statements and is inappropriate. 

minor editing is required. 

Author Response

Thank you for your comments and suggestions. We believe this is a much stronger article with your changes. We have attached our responses to your comments. We are hopeful that our changes meet your expectations.

Reviewer 1

It would be helpful if the abstract contained more data-oriented information and valuable findings.

We have updated the abstract to reflect data-oriented information. The abstract includes identification of key results and a brief statement regarding the importance of the research relative to producer survival in the long-green chile market.

In addition to presenting the data, the paper will also provide more actionable recommendations based on the findings to help U.S. chile producers making the most of the findings.

Please see the updated discussion and conclusion sections.  

There is a need to write down the materials and methods in a section-by-section manner.

We are uncertain of what is meant with this comment or suggestion.  We have divided the Materials and Methods section into data (2.1) and methods (2.2). Given the nature of the research, i.e., social science, a subheading of “data” section better represent the content that “materials.”

I would like to suggest you look into the citation issue, as it is not being used in factual statements and is inappropriate

We had an issue with the formatting template suggested by the journal that caused some issues with the formatting of citations and references that we did not recognize until receiving our review.  We have fixed those issues. In addition, we have attempted to provide more information related to the cited works. In some cases we have provided citations and references to published articles that illustrate the use of a particular method or procedure, e.g., citations 12 and 13 reference commonly used texts that illustrate the importance of discrete choice analysis in consumer preference work.  

Reviewer 2 Report

Thank the authors for this interesting paper. I think there is some room for improvement:

- I miss the purpose of the paper and the main goal is not clear

- the theoretical background is deficient, we need to explain the situation and the studies that have been done according to the mail aim of the paper

- the results need to be explained in more detail, not only in tables

- discussion - it is necessary to add some studies and compare their results with your results

- to conclusions, add some limitations or aren´t there any?

The whole paper needs to be formatted according to the template, references need to be rewritten (not like e.g. Yue, C. &. (2009).)

Author Response

Thank you for your comments and suggestions. We believe this is a much stronger article with your changes. We have attached our responses to your comments. We are hopeful that our changes meet your expectations.

Reviewer 2 

- I miss the purpose of the paper and the main goal is not clear

The purpose of the paper is to better understand U.S. consumer preferences for chile peppers, specifically New Mexico type long-green chile peppers as a means of helping domestic producers survive in an increasingly competitive market.  We have attempted to elaborate on this goal in the introduction, page 3 lines 80 to 87. 

- the theoretical background is deficient, we need to explain the situation and the studies that have been done according to the mail aim of the paper

We believe that the introduction explains the background of the industry and the importance of the research.  In relation to the theoretical background relative to preference work conducted in the realm of peppers, we have included information regarding previous research and have summarized the work in Table 2 (page 6). 

- the results need to be explained in more detail, not only in tables

We have included explanations to supplement the information in the results tables. 

- discussion - it is necessary to add some studies and compare their results with your results

Please see revised discussion section.

- to conclusions, add some limitations or aren´t there any?

We have added limitations and suggestions for further work in the conclusion section.

The whole paper needs to be formatted according to the template, references need to be rewritten (not like e.g. Yue, C. &. (2009).)

We used the template provided by the journal but has several issues with the template that resulting in hidden figures and improperly formatted citations.  These have been fixed in the revised manuscript.

Reviewer 3 Report

This paper utilized a discrete choice experiment to study U.S. consumers’ preferences for pepper attributes. Overall, it is an interesting paper that would provide important information for pepper growers. However, this paper lacks a deep analysis of their data. The mixed logit or latent class model could be used to explore consumer heterogeneity, so that the discussion would be more interesting and objective. Some minor suggestions are as follows:

Line 32 and 89 have an extra space.

Figure 3 is missing.

Line 176 refers to the wrong table number. It should be Table 2 instead of Table 1. Line 192 also refers to the wrong table number. It should be Table 3 instead of Table 2. All table numbers should be rechecked.

Line 176, this table should be placed before Figure 4.

It is necessary to explain why these attributes were chosen.

The conclusion part is too short. Policy implications should be provided.

Author Response

Thank you for your input and suggestions to improve this article. We are hopeful you find the changes impactful and benefitical for the overall effectiveness of this article. 

Reviewer 3

Line 32 and 89 have an extra space.

Please see the significantly revised manuscript.

Figure 3 is missing.

We used the template provided by the journal, but has several issues with the template that resulting in hidden figures and improperly formatted citations.  We apologize for the missing figure.  It was in the manuscript but hidden behind text.  We have attempted to ensure formatting issues have been corrected.

Line 176 refers to the wrong table number. It should be Table 2 instead of Table 1. Line 192 also refers to the wrong table number. It should be Table 3 instead of Table 2. All table numbers should be rechecked.

Please see revised manuscript

Line 176, this table should be placed before Figure 4.

Please see revised manuscript

It is necessary to explain why these attributes were chosen.

Please see revised manuscript.  Explanations for selection are provided in page 5, lines 139 through 143. A summary of previous research related to these attributes used in previous research is provided in Table 2.

The conclusion part is too short. Policy implications should be provided.

We better developed the discussion and conclusions to include implications for stakeholders as well as limitations and potential future research.

Round 2

Reviewer 2 Report

Thank the authors for the revisions. It is acceptable now. Good luck in the next research!

Reviewer 3 Report

The manuscript has been sufficiently improved.